# Greater Height Is Associated with a Larger Carotid Lumen Diameter

**DOI:** 10.3390/medicines6020057

**Published:** 2019-05-14

**Authors:** Phoenix Hwaung, Moonseong Heo, Brianna Bourgeois, Samantha Kennedy, John Shepherd, Steven B. Heymsfield

**Affiliations:** 1Pennington Biomedical Research Center, LSU System, Baton Rouge, LA 70808, USA; phoenix.hwaung@pbrc.edu (P.H.); bbou11@lsuhsc.edu (B.B.); 2Department of Public Health Sciences, Clemson University, Clemson, SC 29631, USA; mheo@clemson.edu (M.H.); Samantha.Kennedy@pbrc.edu (S.K.); 3University of Hawaii Cancer Center, Honolulu, HI 96813, USA; johnshep@hawaii.edu

**Keywords:** cerebrovascular disease, ultrasound, clinical

## Abstract

**Background:** Previous studies link tall stature with a reduced ischemic stroke risk. One theory posits that tall people have larger cerebral artery lumens and therefore have a lower plaque occlusion risk than those who are short. Previous studies have not critically evaluated the associations between height and cerebral artery structure independent of confounding factors. **Methods:** The hypothesis linking stature with cerebral artery lumen size was tested in 231 adults by measuring the associations between height and common carotid artery diameter (CCAD) and intima–media thickness (IMT) after controlling for recognized vascular influencing factors (e.g., adiposity, blood pressure, plasma lipids, etc.). **Results:** Height remained a significant CCAD predictor across all developed multiple regression models. These models predict a ~0.03 mm increase in CCAD for each 1-cm increase in height in this sample. This magnitude of CCAD increase with height represents over a 60% enlargement of the artery’s lumen area across adults varying in stature from short (150 cm) to tall (200 cm). By contrast, IMT was non-significantly correlated with height across all developed regression models. **Conclusions:** People who are tall have a larger absolute CCAD than people who are short, while IMT is independent of stature. These observations potentially add to the growing cardiovascular literature aimed at explaining the lower risk of ischemic strokes in tall people.

## 1. Introduction

There is a longstanding observation that tall people are at a lower risk of developing an ischemic stroke compared to their short counterparts [1,2,3,4,5]. For example, in the 1996 study by Njolstad et al. [4], an inverse dose–response association was observed between height and stroke risk in a sample of middle-aged Finnish men and women. Following adjustment for potentially confounding variables, the risk of stroke was lower by one-half in the tallest quartile of men compared to the shortest quartile of men and by two-thirds in the women. Similarly, McCarron et al. [3] reported in 2000 that subjects in the tallest quartile of their sample had a 30% and 50% lower risk of ischemic and fatal ischemic strokes, respectively, compared to those in the shortest quartile.

The mechanisms reducing the risk of stroke observed in tall people are largely unknown. Hypotheses include height-related hemodynamic [6], genetic [7], and developmental protective factors [8,9]. Reconstruction studies utilizing ex vivo experimentation combined with computational fluid dynamic techniques on the carotid artery [10], iliac artery [11], and aorta [12] may help predict surgical outcomes and provide quantitative insights into the resulting hemodynamic effects from directly altering vessel diameter. Very little information, however, is available on the quantitative relations between stature and cerebral artery lumen areas. One theory suggests that tall people have inherently larger arteries and can thus sustain greater plaque buildup before ischemic effects prevail [13,14]. The common carotid is the most accessible relevant artery for quantitative clinical assessments, although earlier studies linking height to the vessel’s diameter (CCAD) often inadequately controlled for potential confounding factors. To determine whether these variables have any effects on the observed association between height and CCAD, we conducted a prospective examination of CCAD in a sample of adult men and women varying widely in age and adiposity. This study was initiated as an outgrowth of earlier investigations relating blood pressure to height in adults [6,8,9].

## 2. Materials and Methods

The authors agree to make the materials reported in this study available and the senior author, whose contact information is provided, is responsible for maintaining these data sets.

### 2.1. Participants

Adult participants, aged ≥18 years, were recruited as part of the National Institutes of Health Shape Up! Adults research program (R01DK109008). The sample selection was designed to represent a cross-section of the non-institutionalized adult U.S. population. Participants were excluded if they had medical implants, joint replacements, were pregnant, or surpassed the dual-energy x-ray (DXA) scanners weight limit of 200 kg. The study was approved by the Pennington Biomedical Research Center Institutional Review Board. All participants signed an informed consent prior to participation.

### 2.2. Measurements

A minimum overnight fast of 10 h was mandatory for participation. The next morning, a medical evaluation ensured participants met study criteria. The following measurements were made on the same day by trained and certified technicians.

#### 2.2.1. Weight and Height

Participants were barefoot and wore a hospital gown over undergarments. Body weight was measured three times with a digital scale (Holtain Ltd., Crymych, Dyfed, UK) and averaged to the nearest 0.1 kg.

Participants were positioned with their heels, buttocks, and upper back in contact with the stadiometer (Holtain Ltd., Crymych, Dyfed, UK). The slide was lowered to the vertex of the participant’s skull during a breath hold. Three readings were made to the nearest 0.1 cm and averaged. Body mass index (BMI) was calculated as weight/height^2^.

#### 2.2.2. Adiposity

Body composition, including %fat, was measured with the Center’s DXA scanner (Horizon, Hologic, Marlborough, MA, USA). The system was regularly calibrated using the manufacturer’s phantoms and protocol.

#### 2.2.3. Blood Pressure

Systolic blood pressure (SBP) and diastolic blood pressure (DBP) were measured three times and averaged using appropriate cuff sizes with a previously reported standardized protocol [6]. Pulse pressure (PP) was calculated as SBP − DBP. Mean blood pressure (MBP) was calculated as (SBP + 2 × DBP)/3.

#### 2.2.4. Blood Studies

Fasting plasma glucose and lipids were analyzed using a Beckman Coulter UniCel^®^ DxC600 system (Beckman Coulter Inc., Brea, CA, USA) in the center’s certified clinical laboratory.

#### 2.2.5. Ultrasound

Systolic and diastolic right common carotid artery diameter images were obtained in recumbent participants using a 7.5-MHz gel-coated transducer (Aplio 80, Toshiba, Otawara, Japan) and stored on the B-mode system. The system’s caliper tool was used to measure the CCAD and intima–media thickness (IMT) to the nearest 0.1 mm at a horizontal section at least 10 mm in length. CCAD was measured from the intima–lumen interface of the near wall to the far vessel wall. IMT was measured from the lumen–intima to media–adventitia interface of the far vessel wall. An example scan is presented in Appendix A.

### 2.3. Statistical Methods

The statistical analyses were conducted in three stages, the first involving descriptive analyses of sample characteristics. The second stage involved exploratory development of systolic CCAD and IMT multiple regression models. The third stage included the refinement of these models to predict how CCAD and IMT vary with height and other influencing factors.

#### 2.3.1. Exploratory Analyses

Analyses began with the full sample of 231 men and women. Systolic CCAD or IMT were set as dependent variables in multiple regression models with sex, age, height, plasma glucose and lipids, and blood pressure set as potential covariates. To confirm the strength of the initial observations related to stature, we then ran these analyses on a subgroup of 156 participants (93 women) who had no history of chronic disease, were not smokers, and were not taking lipid or blood pressure-lowering medicines. We present these analyses as Models 1 and 2 for CCAD and IMT, respectively. Total cholesterol, MBP, and PP were not included in these models because of potential collinearity.

#### 2.3.2. Main Model Development

Prediction equations were developed describing variation in systolic CCAD and IMT with height as might be observed in the general U.S. population using our full sample of 231 participants. These two models (CCAD, Model 3; IMT, Model 4) were prepared using standard mixed stepwise multiple regression analysis and confirmed using 10-fold cross-validation and cross-validation error (CVE). In Models 5 and 6, we replaced BMI with %fat using the same sample tested in Models 3 and 4.

#### 2.3.3. Sensitivity Analyses

Models 7 through 9 include lipid covariates; therefore, these regression analyses were carried out after excluding participants currently taking lipid-lowering medicines (n = 28). None of the blood pressure covariates entered CCAD or IMT prediction models, and these results are not presented.

All statistical evaluations were conducted in JMP (SAS Institute Inc., Cary, NC, USA) with demographic and other baseline results presented as mean ± standard deviation. Standard errors (SEs) are presented for all of the regression models. The regression model variables were all normally distributed and model interaction terms were non-significant. Between-sex differences in demographic variables were tested using unpaired t-tests and *p* < 0.05 was considered a statistically significant difference. The internal carotid artery area was calculated by assuming the vessel lumen is circular (i.e., area = π(CCAD/2)^2^).

## 3. Results

### 3.1. Descriptive Analyses

The clinical characteristics and medical history of the 231 participants enrolled in the study are shown in Table 1. The sample included 101 men and 130 women ranging in age from 18 to 75 years (mean, 47 ± 17 years) and BMI from 16.0 to 52.6 kg/m^2^ (27.9 ± 6.8 kg/m^2^).

Men and women were similar in age, on average in the fifth decade, and generally overweight (i.e., BMI ≥ 25 kg/m^2^). Overall, women had a significantly smaller mean CCAD, body weight, height, MBP, and DBP, and significantly larger total and high-density lipoprotein-cholesterol, and %fat levels than the men.

### 3.2. Exploratory Analyses

CCAD positively correlated with height before controlling for potential covariates (Figure 1), and was significantly correlated with height after controlling for potential covariates in the full sample (*p* < 10^−5^) of 231 participants and in the smaller sample (*p* = 0.00022) of 156 clinically healthy participants (Model 1, Table 2). BMI (*p* = 0.0053) was also a significant covariate in Model 1. In Model 2 for IMT, only age (*p* = 0.0008) entered as a significant covariate; height (*p* = 0.12) was non-significant, which is also shown in Figure 1.

### 3.3. Prediction Models

The prediction equations developed from the stepwise selection for CCAD (Model 3) and IMT (Model 4) are summarized in Appendix A and were as follows:

CCAD (mm) = −0.258 + 0.029 × Height (cm) + 0.006 × Age (years) + 0.036 × BMI (kg/m^2^)with an *R*^2^ of 0.20 (SE, ± 0.77, *p* < 0.001; CVE, ± 0.04), and(1)

IMT (mm) = 0.25 + 0.005 × Age (years) + 0.005 × BMI (kg/m^2^)with an *R*^2^ of 0.24 (SE, ± 0.18, *p* < 0.001).
(2)

Equation (1) indicates that CCAD increases ~0.03 mm for each 1-cm increase in height; IMT is not influenced by height as indicated by Equation (2).

### 3.4. Sensitivity Analyses

Replacing BMI with %fat in the CCAD model led to sex becoming a significant predictor in place of age; height remained a significant predictor of CCAD and not IMT (Models 5 and 6, Appendix A). Total and low-density lipoprotein-cholesterol entered as individual predictor variables for CCAD among participants who were not taking lipid-lowering medicines. (Models 7–9, Appendix A).

Across all of the developed models (Table 2 and Appendix A) height remained a relatively stable predictor of CCAD of about 0.03 mm/cm and was consistently non-significantly correlated with IMT.

### 3.5. Model Integration

The three main CCAD and IMT predictors presented in Equations (1) and (2) show variable effects of height, age, and BMI on carotid artery structure as descriptively shown in Figure 2. The reference person is 20 years old, of normal weight (BMI, 20 kg/m^2^), and short (150 cm or about 5 ft). Increasing this person’s height to 200 cm, or about 6.5 ft, leads to a 67% increase in the carotid’s internal lumen area but no change in IMT. Increasing age to 70 years leads to a 13% increase in the carotid’s internal lumen area and a 56% increase in IMT. Lastly, moving BMI into the obese range (35 kg/m^2^) increases the carotid’s internal lumen area by 23% and IMT by 17%. Therefore, the effects of variable height on carotid cross-sectional structure are different from those of BMI and age, with the latter two influencing IMT in addition to CCAD.

## 4. Discussion

### 4.1. Height and Carotid Artery Structure

Prompted by reports spanning several decades relating adult height to stroke risk [1,2,3,4,5], we embarked on the current project to test the often-stated contention that tall people have relatively larger cerebral artery diameters than short people. While arteries likely scale in size with height, an intuition confirmed by previous studies [15,16], a critical test of this hypothesis for the clinically accessible CCAD was lacking as other potential lumen size determinants were largely uncontrolled for. Earlier reports focused mainly on well-established anatomic vascular risk factors such as the carotid’s IMT [17,18] and external diameter [16], while others explored associations between lumen diameter and sex [12], age [19,20], BMI [21], or genetic factors [22,23].

The findings reported herein show that the positive association between CCAD and height in adults is so robust that it remains significant even after controlling for age, BMI, adiposity, and other potential covariates including plasma glucose and lipid levels, and brachial artery blood pressures. Additionally, our findings indicate that IMT is not significantly correlated with stature, suggesting that overall vascular characteristics are distinctly influenced by adult height and differ from those of age and BMI, two well-studied major carotid artery structural determinants.

These observations provide additional support to earlier reports relating greater height with larger levels of whole-body oxygen consumption, cardiac output, and left ventricular mass [24,25,26]. People who are tall also have lower brachial artery systolic and pulse pressures after the age of 40 years relative to those who are short, an effect that likely derives from hemodynamic and hydrostatic mechanisms [6,27]. A combination of metabolic, hemodynamic, and hydrostatic effects may account for the observed greater CCAD but unchanged IMT with increasing height. The relative contributions of these multiple stature-related physiological effects to carotid artery structure need to be quantitatively evaluated in future studies.

Clinical observations also link height with coronary and carotid artery pathology and mortality risk. In a widely cited study, Nwasokwa et al. [14] observed a higher prevalence and greater severity of coronary artery disease in short men undergoing cardiac catheterization than in their tall counterparts. O’Conner et al. observed a significantly increased in-hospital mortality risk in patients undergoing coronary artery bypass grafts who had a small left anterior descending coronary artery diameter [28]. Fisher et al. found that small physical size and related small coronary artery diameter predicted post-operative bypass graft mortality, even after controlling for other risk factors [29]. Excess operative mortality in women undergoing coronary artery surgery was postulated by the authors as secondary to their smaller stature and coronary artery diameters. Small body size, as defined by height, is a risk factor for perioperative stroke and death in women undergoing carotid endarterectomy [13].

### 4.2. Height and Other Vascular Structures

Previous studies report significant height associations with the diameters of other vascular structures such as the aorta [30,31,32], left ventricle [33], common femoral artery [34,35], main pulmonary artery [36], mid-left descending coronary artery [14,28], and popliteal artery [37]. The effects of height on vascular structure thus go well beyond the carotid artery with stature appearing to influence multiple cardiovascular system components. By inference, cerebral arteries distal to the common carotid are also larger in people who are tall than in those who are short.

### 4.3. Age and Adiposity Effects on Carotid Structure

This study confirms the positive correlation between age and IMT observed in previous studies [38]. The effects of adult age on CCAD reported in previous studies are less clear with responses ranging from an increase [19,20] to no change [39] in lumen diameter, discrepant observations that may be due to differences in participant sample size or age range. The reported increase in CCAD with greater age may be secondary to endogenous physiological mechanisms that favor enlargement of the arterial lumen [20,40,41]. In the current study, we observed a significant correlation between age and CCAD after controlling for height and BMI, although this association weakened in our clinically healthy subgroup.

As with other reports, we also confirmed an increase in both CCAD and IMT as a function of BMI [17,42,43], and more specifically, adiposity as defined by %fat. Kappus et al. attributed these vascular effects to an adiposity-related increase in blood pressure, notably that of the carotid artery [21,44]. In a previous study, we also observed a significant correlation between mean brachial artery blood pressure and BMI in non-hypertensive adults in the USA population [6]. Importantly, in the current study, height remained a significant predictor of CCAD, independent of both age and BMI in all of the models tested.

### 4.4. Study Limitations

Our sample size was relatively small, particularly when we excluded participants in subgroup analyses. While our findings related to height and CCAD were robust, we cannot exclude subtle covariate effects (e.g., race, physical activity level, etc.) or interaction effects that might be discerned with greater study power. We were also unable to directly measure local carotid blood pressure, which previous studies suggest may be a better predictor of carotid physiology compared to peripheral blood pressure measurements [38,45]. Additionally, we made the assumption that the cross-section of the carotid artery is circular in shape. Analyzing the associations between height and carotid artery blood velocity, blood viscosity, or elasticity may help to elucidate possible mechanisms through which stature affects vascular properties.

## 5. Conclusions

In conclusion, we critically tested the hypothesis that the carotid artery’s luminal diameter is significantly associated with adult stature. This theory was prompted by the often stated but not critically tested assertion that tall people are at a lower risk of experiencing an occlusive ischemic stroke than people who are short because they have larger cerebral artery lumens. We confirmed this hypothesis, demonstrating that the carotid’s lumen increases with height. These findings suggest that arteries with larger lumens in general supply the central nervous system in people who are tall relative to those who are short. These and related blood pressure and clinical observations [6] suggest that height plays a more important role in cardiovascular physiology and pathology than previously recognized.

## Figures and Tables

**Figure 1 medicines-06-00057-f001:**
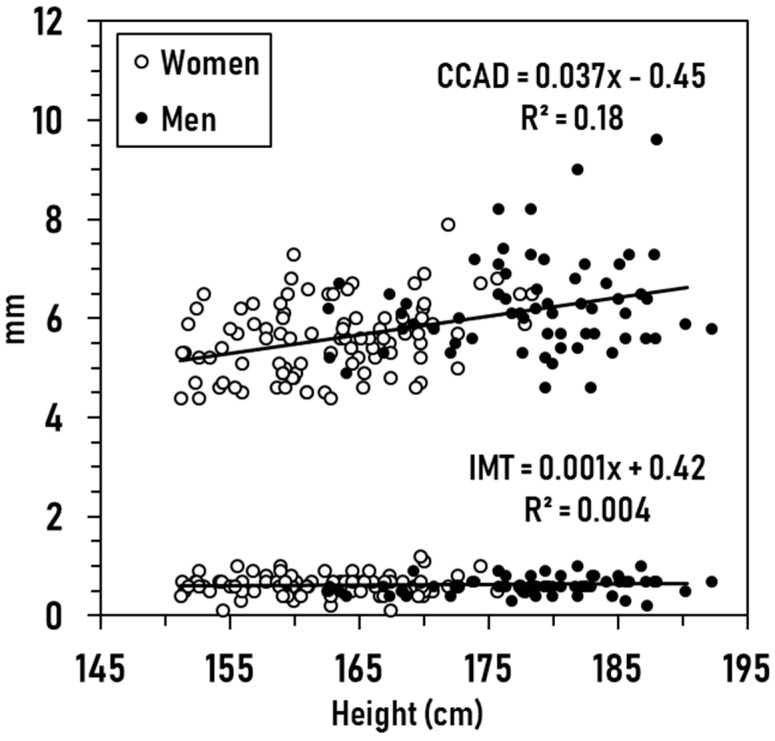
Measured CCAD (mm) and IMT (mm) in men and women versus height (cm). The CCAD correlation is significant at *p* < 0.001, while the IMT correlation is non-significant (*p* = 0.41).

**Figure 2 medicines-06-00057-f002:**
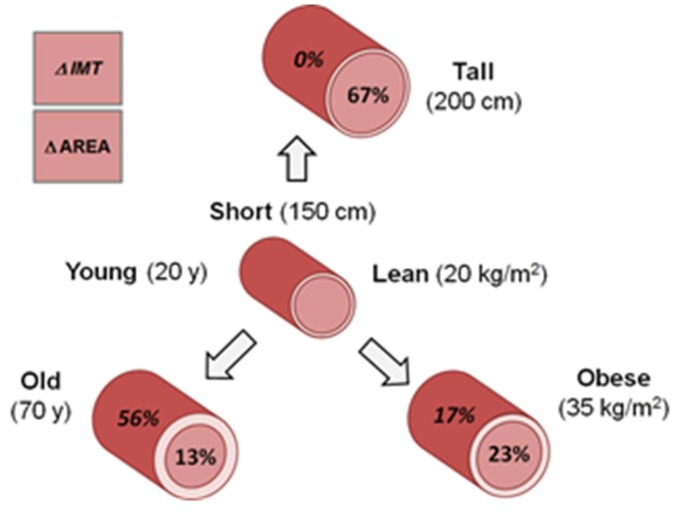
Hypothetical cross-sections of the common carotid artery showing the estimated percentage differences in internal lumen areas and intima–media thicknesses between a short, young, normal (Nl) weight reference person and their tall, old, and obese counterparts. Calculations are based on respective CCAD and IMT prediction equations #1 and #2 in the text and Appendix A.

**Table 1 medicines-06-00057-t001:** Subject characteristics.

	Men	Women	Total
**N**	101	130	231
**Age (years)**	45.7 ± 17.7	47.7 ± 17.0	46.8 ± 17.3
**Weight (kg)**	90.2 ± 21.8	72.3 ± 19.2 ^‡^	80.1 ± 22.2
**Height (cm)**	177.3 ± 6.9	162.4 ± 6.6 ^‡^	168.9 ± 10.0
**BMI (kg/m^2^)**	28.6 ± 6.3	27.4 ± 7.1	27.9 ± 6.8
**%Fat**	26.3 ± 6.3	37.9 ± 7.5 ^‡^	32.9 ± 9.1
**CCAD (mm)**	6.16 ± 0.88	5.72 ± 0.79 ^‡^	5.92 ± 0.85
**IMT (mm)**	0.66 ± 0.20	0.64 ± 0.20	0.65 ± 0.20
**Plasma Glucose (mg/dL)**	96.3 ± 20.2	94.7 ± 18.8	95.4 ± 19.4
**TG (mg/dL)**	97.5 ± 63.0	86.6 ± 46.6	91.32 ± 54.5
**LDL (mg/dL )**	104.9 ± 30.1	112.7 ± 30.3	109.3 ± 30.4
**HDL (mg/dL)**	53.4 ± 14.3	63.5 ± 16.7 ^‡^	59.1 ± 16.5
**TC (mg/dL)**	177.8 ± 37.4	193.8 ± 38.3 ^‡^	186.8 ± 38.7
**SBP (mmHg)**	119.2 ± 12.8	116.6 ± 15.2	117.7 ± 14.2
**DBP (mmHg)**	77.9 ± 8.3	73.9 ± 8.8 ^‡^	75.6 ± 8.8
**PP (mmHg)**	41.3 ± 9.6	42.7 ± 11.4	42.1 ± 10.7
**MBP (mmHg)**	91.6 ± 0.89	88.1 ± 0.87 ^†^	89.7 ± 0.64
**Medical History**			**Total**
**Blood Pressure (BP)**			47
**Only BP**			22
**+Chol**			11
**+Diabetes**			8
**+Chol and Diabetes**			4
**Cholesterol (Chol)**			28
**Only Chol**			8
**+Diabetes**			2
**Type II Diabetes**			26
**Only Diabetes**			7
**Heart Disease**			5
**Only Heart Disease**			0
**+Diabetes**			3
**+BP and Chol**			1
**+Diabetes, BP, and Chol**			1
**Current Smokers**			17
**Only Smokers**			15
**+BP**			1
**+Diabetes**			1

Characteristics of the total sample of men and women, expressed as X ± SD. Abbreviations: BP, participants taking blood pressure medication; BMI, body mass index; CCAD, common carotid artery diameter in systole; Chol, participants taking cholesterol-lowering medications; DBP, diastolic blood pressure; HDL, high-density lipoprotein cholesterol; IMT, carotid intima–media thickness in systole; LDL, low-density lipoprotein cholesterol; MBP, mean blood pressure; PP, pulse pressure; SBP, systolic blood pressure; TC, total cholesterol; plasma TG, triglycerides. ^†^, *p* < 0.01; ^‡^, *p* < 0.001 versus men.

**Table 2 medicines-06-00057-t002:** Right common carotid artery diameter (CCAD) and intima–media thickness (IMT) multiple regression models (n = 156).

	Model 1: CCAD			Model 2: IMT		
**Covariate**	**ß**	**95% CI**	**SE**	**ß**	**95% CI**	**SE**
**Intercept**	−0.62	−4.25 to 3.00	1.84	−0.29	−1.09 to 0.51	0.41
**Height (cm)**	0.035 ^‡^	0.017 to 0.053	0.009	0.003	−0.001 to 0.007	0.002
**BMI (kg/m^2^)**	0.047 ^†^	0.014 to 0.080	0.017	0.004	−0.003 to 0.011	0.004
**Age (years)**	0.007	−0.003 to 0.016	0.005	0.004 ^‡^	0.002 to 0.006	0.001
**Sex (0, F; 1, M)**	0.172	−0.40 to 0.74	0.289	-0.041	−0.167 to 0.086	0.064
**%Fat**	0.009	−0.020 to 0.039	0.015	0.0002	−0.006 to 0.007	0.003
**PGlu (mg/dL)**	−0.004	−0.022 to 0.014	0.009	0.002	−0.002 to 0.006	0.002
**TG (mg/dL)**	0.001	−0.002 to 0.004	0.001	0.00008	−0.0005 to 0.0006	0.0003
**LDL (mg/dL)**	−0.005	−0.010 to 0.0001	0.003	0.0003	−0.001 to 0.001	0.0006
**HDL (mg/dL)**	0.005	−0.004 to 0.013	0.004	−0.0003	−0.002 to 0.002	0.001
**SBP (mmHg)**	−0.002	−0.016 to 0.011	0.007	0.0003	−0.003 to 0.003	0.002
**DBP (mmHg)**	−0.008	−0.029 to 0.013	0.011	−0.002	−0.007 to 0.003	0.002
***R*^2^**	0.27 ^‡^		0.77	0.13 ^‡^		0.17

^†^, *p* < 0.01; ^‡^, *p* < 0.001. BMI, body mass index; CI, 95% confidence interval; DBP, diastolic blood pressure; HDL, high-density lipoprotein cholesterol; IMT, carotid intima–media thickness in systole; LDL, low-density lipoprotein cholesterol; PGlu, plasma glucose; SBP, systolic blood pressure; SE, standard error.

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
