# Peer review of "Greater Height Is Associated with a Larger Carotid Lumen Diameter"

_medicines, 2019, doi:10.3390/medicines6020057_

Round 1
Reviewer 1 Report
Attached

Reviewer 2 Report
1. In section 3.2, when mentioning the statistically significant values, could you state the actual p-values in the text for perspective (unless they are less than 10-5). Also, regarding the result that height was not a significant factor in the IMT model, could you state the actual p-value in the text for reference.
2. As a supplementary table, can you have a repeat of Table 1 with the cohort of 156 clinically health participants subjects (to help show to what extent the healthy cohort is representative of the set of all subjects)? Alternatively, please state (around line 141) the breakdown of male vs female in the healthy cohort.
3. In section 3.3 Prediction Models, the SE, p, and CVE ended up lumped in with equation 1. Even though this information is in Table S1, it’d be nice to have this reported in the main text for Equation 2 as well.
4. It would be illustrative to show a scatter plot with confidence bands of the actual vs predicted CCAD and IMT based on equations 1 and 2. This also helps illustrate the range of CCAD and IMT measurements collected in the study. It could also be nice to represent the healthy cohort vs the rest of the subjects with different markers to get a sense of overlap, although this might be confusing to try and interpret. Alternatively, it could also be illustrative to represent male vs female with different markers in this plot.
5. It would be helpful to have a bit more detailed discussion comparing and contrasting previous specific results, in particular from Krejza et al 2006 or Polak et al. 1996. This could help clarify to what extent specific prior findings were or were not confirmed in this study data (albeit with some specific differences in parameters gathered) and also help highlight the differences in approach.
Round 2
Reviewer 1 Report
I am satisfied with the author's correcions